# Integrated Episodic and Semantic Memory via Modulating Transformer FeedForward Layers

Yiqun Yao [1 2]  Xiang Li [1 2]  Xin Jiang [1]  Xuezhi Fang [1]  Naitong Yu [1]  Siwei Dong [1]  Wenjia Ma [2]  Jing Li [3]
Aixin Sun [4]  Yequan Wang [1 2]

## Abstract

It is widely recognized that, after generative pre-training, Transformer FeedForward layers implicitly function as semantic memory, encoding linguistic and factual knowledge, while the contexts in key–value (KV) cache contain raw events, serving as the source of models' episodic memory. In this work, we show that a same group of Transformer FeedForward-layer parameters can both be semantic and episodic memory, which is retrievable without explicitly attending to the related KV cache. To realize this idea, we introduce HyperMem, a hypernetwork that recurrently maps contexts into targeted updates of FeedForward parameters. We post-train the hypernetwork using continuation and random-access associative memory objectives, eliminating the need for test-time training. Extensive experiments demonstrate that our approach outperforms related methods, including MemoryLLM and generative adapter, on memory retrieval, long-context question answering, and personalization benchmarks, establishing a new state of the art for hypernetwork-based memory mechanisms. Our results suggest that directly bridging data and parameters provides a viable direction for exploring next-generation foundation models with more flexible and persistent memory capabilities.

## 1. Introduction

The progress toward Artificial General Intelligence (AGI) (Bubeck et al., 2023) has been significantly advanced by

[1]Beijing Academy of Artificial Intelligence, Beijing, China [2]Spin Matrix, Beijing, China [3]Harbin Institute of Technology, Shenzhen, China [4]Nanyang Technological University, Singapore. Correspondence to: Yequan Wang <tshwangyequan@gmail.com>.

*Proceedings of the 43rd International Conference on Machine Learning*, Seoul, South Korea. PMLR 306, 2026. Copyright 2026 by the author(s).

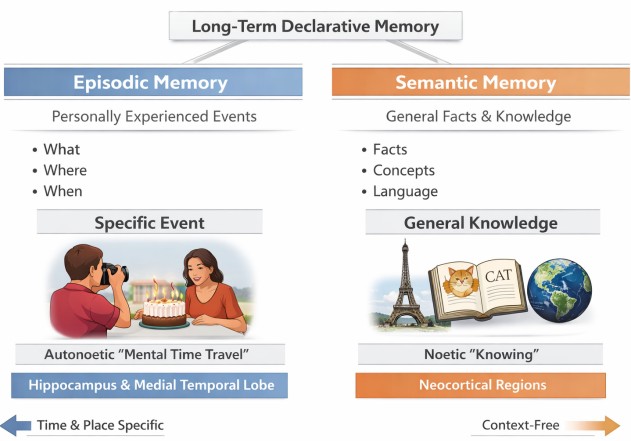

*Figure 1.* **Episodic memory vs. semantic memory: definition**.

large language models (LLMs) (Hurst et al., 2024; Bai et al., 2022; Kaplan et al., 2020). When combined with reinforcement learning (Schulman et al., 2017; Rafailov et al., 2023) and test-time scaling laws (Jaech et al., 2024), contemporary LLMs are now capable of solving a wide range of complex problems, including programming and mathematics (Lightman et al., 2023). Nevertheless, a defining characteristic of human intelligence lies in its ability to continuously acquire, retain, and adapt knowledge from a lifelong stream of experiences (Tulving, 1985; Atkinson & Shiffrin, 1968). In contrast, current LLMs still lack robust memory capabilities (Wang et al., 2024c). As a result, incorporating memory has become a central challenge in the design of next-generation foundation models and learning algorithms (Behrouz et al., 2024).

Human memory can be categorized along multiple dimensions. From a temporal perspective, it consists of short-term and long-term memory (Atkinson & Shiffrin, 1968); from a functional perspective, it includes episodic, semantic, and procedural memory (Tulving, 1985). In this paper, we focus on implementing two core functions in LLMs: *episodic memory*, the capability to recover textual "events" tied to hints on time/position, and *semantic memory*, which is the ever-updating knowledge and language modeling capability (Figure 1). In modern LLMs, episodic memory is typi-

cally converted from textual contexts (working memory) summarized by retrieval-augmented generation (RAG) or agent-based systems. Semantic memory, by contrast, is generally regarded as primarily encoded in model parameters. Prior work has provided empirical evidence that FeedForward (MLP) layers in Transformer architectures function as a form of key–value semantic memory (Geva et al., 2021).

Intuitively, episodic and semantic memory are not isolated but can be transformed into one another. In humans, this transformation is achieved through mechanisms such as replay and memory reconsolidation (Wilson & McNaughton, 1994; Squire & Alvarez, 1995). For models, however, existing approaches remain limited. Knowledge editing techniques (Wang et al., 2024a) typically update factual knowledge while neglecting fine-grained episodic details and reasoning abilities, whereas test-time training (Wang et al., a;b) is often constrained by latency and computational overhead. In this work, we explore an alternative perspective: dynamically converting episodic memory (data, kv cache, etc.) into semantic memory (parameters). This design offers two key advantages. First, inference no longer relies on quadratic-cost context windows or test-time learning, resulting in faster response times and reduced memory overhead. Second, the same data can be preserved both as intact episodic memory and as distilled semantic memory, in the same group of parameters, with switching achieved solely through prompting.

To realize this idea, we introduce HyperMem, a hypernetwork that maps inference-time contexts directly into parameter modulation of FeedForward layers. Specifically, (1) a trainable module incrementally updates a fixed-size memory pool at each Transformer layer upon each data chunk, and (2) at any time, a hypernetwork projects this memory pool into FeedForward-layer parameters. These projected parameters are added to the frozen base model weights, acting as a learned modulation mechanism. We train the hypernetwork with two major objectives: one encourages the MLP layers to reproduce *episodic* context through associative memory mechanism, while the other promotes the *semantic* integration of acquired knowledge for subsequent tasks.

Experimental results demonstrate that our model exhibits integrated episodic and semantic memory capabilities even in context-free settings. In particular, its raw event retrieval capabilities significantly surpasses prior methods such as MemoryLLM (Wang et al., 2024b) and generative adapter (Chen et al., 2025). On semantic understanding benchmarks such as LongBench (Bai et al., 2024), our model achieves performance comparable to approaches that rely on fixed-size external memory. Overall, our work bridges episodic experience and semantic knowledge within a unified model representation.

## 2. HyperMem: Method

HyperMem, our method, is presented in Figure 2. It dynamically encodes contextual information into a fixed-size memory while reading input tokens. Unlike conventional approaches that accumulate memory by extending the context window, HyperMem maintains a recurrent memory pool at each Transformer layer, whose size remains constant. During inference, this memory pool is *not* attended to as part of the input sequence. Instead, it is decompressed into parameter matrices with shapes align with the *up*, *down*, and *gate* projections of the FeedForward network (FFN), and is used to modulate the frozen backbone parameters.

### 2.1. Memory Injection

We refer to the process of dynamically updating memory conditioned on contextual inputs as **memory injection**. Given a chunk of hidden states $c_t = \{h_1, \ldots, h_{\text{seq\_len}^t}\}$, the memory is updated via a memory management function $f$:

$$M_t = f(c_t, M_{t-1}). \tag{1}$$

**Layer-wise Recurrent Memory.** HyperMem maintains a fixed-size memory pool at each Transformer layer. Each pool only receives hidden states from the corresponding layer and does not directly consume hidden states or memory pools from other layers. Formally, for each layer $L$, the memory pool consists of $K = 3r$ slots, where $r \in \mathbb{Z}$, and shares the same dimensionality $d$ as the model hidden states:

$$M_t^L = \{m_1, \ldots, m_K\}^L \in \mathbb{R}^{K \times d}. \tag{2}$$

**Management Function.** At each injection step, the contextual hidden states $c_t^L$ are concatenated with the memory from the previous step $M_{t-1}^L$ along the temporal dimension and passed through the Transformer layer. The resulting hidden states are propagated to the next *layer*, while the updated memory is fed into the next injection *step* (Figure 2, left). All backbone Transformer parameters are frozen. We introduce trainable LoRA (Hu et al., 2022) adapter on the attention $q$, $k$, $v$ projections and the FFN to implement the management function $f$. Thus, equation 2 translates into:

$$c_t^{L+1}, M_t^L = \text{LoRA}_{\text{enc}}(c_t^L, M_{t-1}^L). \tag{3}$$

During memory injection, we modify the original causal attention mask (Meta, 2024) of the backbone model to full bidirectional attention (Devlin et al., 2019) over the concatenated sequence $(c_t^L, M_{t-1}^L)$.

### 2.2. HyperNetwork

After any number of injection steps, the memory pool can be transformed into parameter modulations that participate

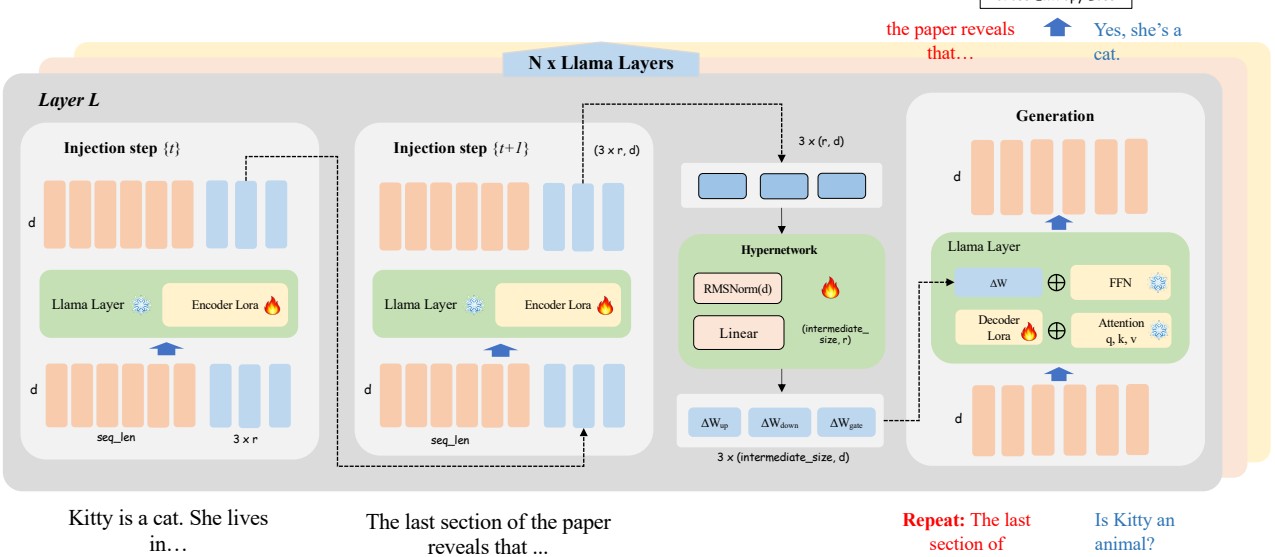

*Figure 2.* **Model structure and computation flow of HyperMem.**

directly in generation, enabling the model to exhibit factual knowledge and language modeling capabilities derived from the injected context. This transformation is implemented by a layer-wise hypernetwork:

$$\Delta W_{\text{up}}^L = \text{HyperNet}_{\text{up}}^L(M_{t,0:r}^L), \quad (4)$$

$$\Delta W_{\text{down}}^L = \text{HyperNet}_{\text{down}}^L M_{t,r:2r}^L), \quad (5)$$

$$\Delta W_{\text{gate}}^L = \text{HyperNet}_{\text{gate}}^L(M_{t,2r:K}^L). \quad (6)$$

The HyperNet consists of an RMSNorm ([Zhang & Sennrich, 2019](#)) layer with hidden dimension $d$, followed by a linear projection from input dimension $r$ to output dimension intermediate_size, which matches the dimensionality of the FFN intermediate layer (e.g., approximately $d \times \frac{8}{3}$ in LLaMA-style ([Touvron et al., 2023](#)) architectures). The resulting tensors $\{\Delta W_{\text{up}}, \Delta W_{\text{down}}, \Delta W_{\text{gate}}\}^L$ are added to the corresponding frozen FFN parameters $\{W_{\text{up}}, W_{\text{down}}, W_{\text{gate}}\}$.

**Connection to LoRA.** From a computational perspective, HyperMem is closely related to LoRA with rank $r$ applied to the FFN projection matrices. Taking $W_{\text{up}}$ as an example, let $A = \text{RMSNorm}(M_{0:r}) \in \mathbb{R}^{d \times r}$ and let $B \in \mathbb{R}^{r \times \text{intermediate\_size}}$ denote the projection matrix of the HyperNet. The resulting modulation satisfies:

$$\text{rank}(\Delta W_{\text{up}}) = \text{rank}(AB) \leq \min(\text{rank}(A), \text{rank}(B)) \leq r \quad (7)$$

Unlike conventional LoRA with directly-trainable parameters, HyperMem can be viewed as a *hyper-LoRA*, where the left factor $A$ is dynamically generated from the contextual memory, and the right factor $B$ is a learned conversion

function. HyperMem is also special in that the bottleneck dimension $r$ directly interacts with the *temporal* dimension of the Transformer layer input, encouraging the management function $f$ to preserve language modeling capabilities while incorporating gradients propagated from the parameter space.

### 2.3. Generation

During generation, the transformed parameters are used to modulate the FFN layers with a multiplier $\alpha$ [1]:

$$W_{\text{pos}}^* = W_{\text{pos}} + \alpha \Delta W_{\text{pos}}, \quad \text{pos} \in \{\text{up}, \text{down}, \text{gate}\}, \quad (8)$$

$$\text{FFN}(h) = W_{\text{down}}^* \big((W_{\text{up}}^* h) \odot \text{act\_fn}(W_{\text{gate}}^* h)\big), \quad (9)$$

in which act_fn is the activation function from the backbone. To adapt to the modulated FFN layers, we further leverage a trainable decoder-side LoRA module on the attention $q$, $k$, and $v$ projections during decoding. This $\text{LoRA}_{\text{dec}}$ is applied on exactly the same backbone as $\text{LoRA}_{\text{enc}}$, activated in different phases (Figure 2, right).

**Context-free Decoding.** To evaluate the feasibility of parameterized memory, we perform decoding without including any injected text in the input context. Instead, we use prefix prompts to elicit either episodic or semantic memory capabilities, with both behaviors triggered solely by different prompts. A beneficial side effect of this design is the elimination of the $O(N^2)$ attention cost associated with long context windows. Nevertheless, we valid in Section 4.2 that this mechanism is a "plug-in" that does not severely

---

[1] $\alpha$ is a hyperparameter tuned with validation results

harm the model's normal use of the full context window.

## 3. Training Strategy

We follow the training protocol of MemoryLLM (Wang et al., 2024b) and its subsequent extension M+ (Wang et al., 2025a), and adopt LLAMA-3.1-8B (Meta, 2024) as the frozen foundation model. HyperMem is trained in two stages, *post-training* and *fine-tuning*, to bridge the boundary between data and parameters and to integrate episodic and semantic memory within a single model.

### 3.1. Model Configurations

Both the $\mathrm{LoRA_{enc}}$ and $\mathrm{LoRA_{dec}}$ use a rank of 64, scaling factor $\alpha = 128$, and LoRA dropout of 0.1. The $\mathrm{LoRA_{enc}}$ is applied to the attention $q$, $k$, $v$ projections as well as the FFN $up$, $down$, and $gate$ matrices, whereas the $\mathrm{LoRA_{dec}}$ is applied only to the attention projections.

Each Transformer layer maintains a memory pool of size $K = 3 \times 86, d = 4096$, whose initial state ($t = 0$) is a set of trainable parameters. We additionally introduce a trainable beginning-of-sequence (BOS) embedding and a special positional embedding to mark memory slots, following the design choices from MemoryLLM-8B (Wang et al., 2025a).

### 3.2. Post-training

**Objectives.** Our goal in post-training is to encourage a single set of FFN parameters to support both episodic reconstruction of contextual details and semantic utilization of knowledge, thereby achieving integrated memory. To this end, we design two complementary objectives.

*(1) Random-access associative memory.* After injecting contextual text chunks, we prepend the prefix prompt ``Repeat:  " followed by five randomly selected tokens from the original text, and task the model with recovering the corresponding continuation. This objective encourages the model to retain fine-grained episodic information. The prefix tokens play the role of "spatiotemporal hints" in episodic memory, tailored for the textual modality.

*(2) Continuation.* Given several consecutive chunks from the same document, the model is required to generate the subsequent chunk. In this setting, the prefix prompt is empty and generation starts from a BOS token. This objective promotes semantic abstraction and language modeling capability over injected content.

**Data.** We use a subset of the FineWeb-Edu (Penedo et al., 2024) corpus for post-training. Documents are segmented into chunks with randomly sampled lengths between 16 and 512 tokens. Ten percent of the data is reserved for standard causal language modeling without memory injection.

The remaining data is evenly split between the two objectives described above. Training is conducted with batch size 32 for one epoch, corresponding to approximately 93k optimization steps.

**Learning Configurations.** To balance GPU memory consumption and training efficiency, we limit gradient backpropagation to at most 13 injection steps; earlier injections are detached from the computation graph. We use a peak learning rate of $1.6 \times 10^{-4}$ with linear warmup, followed by linear decay to $1.6 \times 10^{-5}$ over the course of one epoch. This strategy stabilizes optimization while enabling effective gradient flow from parameter-space objectives.

### 3.3. Fine-tuning

During fine-tuning, we sample reading comprehension datasets with moderately long contexts ranging from 8k to 16k tokens. Dataset statistics are summarized in Table 1. For each sample, a short suffix of the document together with task instructions is used as the prefix prompt, while all preceding text is injected into model parameters via Hyper-Mem.

*Table 1.* SFT data collection.

| Source | Samples |
|---|---|
| LongAlpaca (Chen et al., 2023) | 12000 |
| Booksum (Kryściński et al., 2022) | 4889 |
| Synthetic books&papers (Zhang et al., 2025) | 10000 |
| GA SFT collection (Chen et al., 2025) | 33631 |
| SlimPajama-long (Shen et al., 2023) | 1000 |
| FineWeb post-training data (Section 3.2) | 8000 |

Fine-tuning uses only the *continuation* objective except on the FineWeb data. We adopt an initial learning rate of $4 \times 10^{-5}$, linearly decaying to $5 \times 10^{-6}$ over 10k steps. To support longer gradient propagation through memory injections, we enable activation checkpointing (`torch.utils.checkpoint`), allowing gradients to flow through approximately 5k injected tokens at the cost of reduced training speed.

## 4. Experimental Results

The core hypothesis of this work is that modulating FFN-layer parameters enables a unified representation of episodic and semantic memory. Accordingly, our experiments focus on evaluating the effectiveness and the efficiency of the trained hypernetwork in realizing these capabilities. In Section 4.1, we evaluate episodic memory, measuring retrieval accuracy through a raw text retrieval task, as well as assessing memory retention under multiple injection steps. In Section 4.2, we examine semantic memory through context-independent language modeling, reading comprehension,

*Table 2.* Raw text retrieval evaluation.

| Model | Equivalent Storage Size | F1: From Head | F1: Random Access |
|---|---|---|---|
| MemoryLLM-8B-12800 (Wang et al., 2025a) | 12800 | 0.901 | 0.645 |
| MemoryLLM-8B-1024 (Wang et al., 2025a) | 1024 | 0.749 | 0.505 |
| Generative Adapter (Chen et al., 2025) | 256 | 0.390 | 0.297 |
| HyperMem | 258 | **0.876** | **0.654** |

and personalization tasks, which simulate scenarios where textual experience is transformed into semantic knowledge. Finally, Section 4.3 presents ablation studies on critical design choices.

### 4.1. Episodic Memory

Recall that we define *episodic memory* as the capability to recover textual "events" tied to hints on time/position (Section 1). To evaluate episodic memory, we randomly sample documents from a FineWeb-Edu subset containing 50,000 documents which we held out for validation. We construct evaluation using documents with length exceeding 550 tokens. Unless otherwise specified, long documents are segmented into chunks of 512 tokens and injected sequentially.

In all episodic memory experiments, we compare HyperMem with MemoryLLM-8B (Wang et al., 2025a), which uses the same backbone model and training corpus as our post-trained model. MemoryLLM-8B represents a context-based memory approach with explicit fixed (yet large) number of memory tokens. We compare with two configurations of MemoryLLM-8B using 1024 and 12800 memory pool tokens, depending on task, while our "memory pool" can be considered as only $86 \times 3 = 258$ tokens. We additionally include Generative adapter (Chen et al., 2025) as a state-of-the-art context-free memory baseline, which features additional storage equivalent to 256 tokens.

#### 4.1.1. RAW TEXT RETRIEVAL

We first measure to what extent the models recalls certain episodic events based on "spatiotemporal" hints through the raw text retrieval task. In this task, the full document is first injected into the model. The model then receives an instruction prompt consisting of the prefix ``Repeat:" concatenated with five continuous tokens starting from the head or a random position in the original document. The model generates 128 tokens, which are decoded, normalized, and compared against the corresponding ground-truth continuation from the original text. Retrieval accuracy is measured using F1 score, a metric widely adopted in benchmarks such as LongBench (Bai et al., 2024) and shown to correlate well with event-level similarity.

Quantitative results are reported in Table 2. HyperMem achieves substantially stronger retrieval performance than related methods while requiring fewer memory tokens and comparable computational overhead. We observe that all models are severely affected by the random position of hints, indicating that episodic memory remains a challenging task, especially for context-free memory models. HyperMem demonstrates significantly better robustness in random access compared to Generative Adapter. One possible explanation could be that Generative adapter suffers from an additive state design (Chen et al., 2025) that weakens their ability to perform random-access retrieval, leading to frequent hallucinations, while our method intrinsically replaces the "additive pooling" of memory states with "recurrent Transformers" which alleviates the problem.

**Case Studies.** We provide a qualitative failure case study in Table 3. Interestingly, in most cases, models fail with the pattern of "generating correct sequence for some steps, and suddenly loses track", as showcased in the example from MemoryLLM. However, we observe in some cases where HyperMem can recover from wrong details (e.g, "input lag" vs. "lead delay" in our case), and keep sticking to the current event rather than drifting into unrelated content. This retrieval pattern appears unique to such "parameter-based episodic memory" and resembles associative memory behaviors observed in humans, as memory recall is widely regarded as *generative and reconstructive* rather than *verbatim*.

#### 4.1.2. RETENTION CURVE

To probe the limits of memory retention imposed by architectural design, we conduct a controlled experiment using a single objective: verbatim text repetition from the document head, on the same FineWeb corpus. As a baseline, we train a MemoryLLM model with memory size $4 \times 256 = 1024$, and compare it against HyperMem with memory size $3 \times 86$. We measure perplexity (PPL) of the recovered text under increasing numbers of injection steps.

Results are shown in Figure 3. HyperMem maintains stable retention even beyond 20 injection steps, while its nominal memory token length is only 258. MemoryLLM exhibits rapid degradation as the number of injections increases. This

*Table 3.* Case Study demonstrating the unique failure pattern of HyperMem. Any incorrect word is colored red.

| Prompt | Repeat: September 13th, 2011 Written |
|---|---|
| Ground Truth | By: Mike Zaimont Yo – let's talk about GGPO. What is this "lag?" Lag is any extra delay in character response beyond what is normal for the game. "Input lag" in game design terms refers to any amount of time that elapses between a player's input and the corresponding character response. All games have some, and in fact the smallest achievable amount of lag is 3 frames at 60 FPS (an explanation of which is beyond the scope of this post). |
| MemoryLLM-8B | By: Mike Zaimont Yo – let's talk about GGPO. What is this "lag?" Lag is any extra delay in character response beyond what is normal, and this is something that all games have, even offline. For example, in Fight: Game INJURY MAXIMUM 80:000 |
| HyperMem | By: Mike Zaimont (missing) What is this "lag?" Lag is any extra delay in character response beyond what is normal for the game. "Lead delay" in game design terms refers to any amount of time that elapses between a player's input and the corresponding character response. All games have some, and in fact the game delays the time it takes players to turn on |

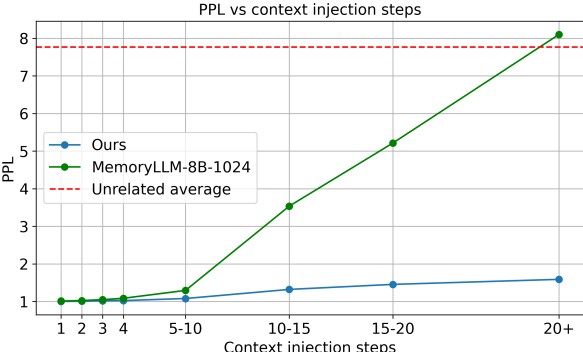

*Figure 3.* **Retention curve: HyperMem vs. MemoryLLM with similar scale of memory tokens**.

difference arises from fundamental design choices. MemoryLLM maintains a fixed-length memory pool for attention and propagates gradients only through the single most recent injection step. Such context-based method exhibits very high precision when the queried content lies within its effective memory scope, but might be less effective as the injected context exceeds the capacity. In contrast, HyperMem allows gradients to flow across multiple injection steps during training, enabling the model to retain information over longer temporal horizons. These results suggest that parameter-based memory modulation offers intrinsic advantages in compression rate for long-range episodic retention.

## 4.2. Semantic Memory

We define *semantic memory* as the model's ability to answer comprehension-oriented and semantic questions about injected context after the data has been transformed into parameter modulations, rather than merely reproducing raw textual events. This setting imposes a stronger requirement on the modulated parameters: it is well known that much of a Transformer's reasoning capability relies on its self-attention module to retrieve relevant contextual tokens, whereas context-free reasoning inherently lack this advantage. To measure this capability, we fine-tune the model and evaluate it on tasks including context-independent language modeling, reading comprehension, and personalization.

### 4.2.1. CONTEXT-INDEPENDENT LANGUAGE MODELING

Although our model does not rely on context to store memory, it is crucial to ensure that *language modeling performance on text unrelated to the injected episodic memory is not degraded when FFN parameters are already modulated*. This property is necessary for models equipped with hypernetwork-based memory to function as vanilla language models when contextual attention is required or allowed.

We evaluate this behavior on the FineWebEdu validation set by measuring average perplexity on unrelated text *with* and *without* memory injection. The resulting perplexities are **7.77** and **7.66**, respectively, demonstrating negligible degradation. These results indicate that parameter modulation via HyperMem is compatible with existing language modeling behavior and does not interfere with the model's ability to process unrelated contextual inputs.

### 4.2.2. READING COMPREHENSION

Following the MemoryLLM line of work (Wang et al., 2024b; 2025a), we evaluate reading comprehension performance on the LongBench benchmark (Bai et al., 2024) using six subtasks: HotpotQA (Yang et al., 2018), NarrativeQA (Kočiskỳ et al., 2018), Qasper (Dasigi et al., 2021), Multi-FieldQA (Bai et al., 2024), 2WikiMQA (Ho et al., 2020), and MuSiQue (Trivedi et al., 2022). Results are reported in Table 4. We observe substantial variance across models and tasks, which can be attributed to differences in training objectives and prompt formats. Nevertheless, HyperMem achieves significantly stronger overall performance compared to generative adapter with the same storage size, even comparable to the full-context MemoryLLM with 12800 tokens, which is 3x slower in inference than its 1024-token version. This suggests that parameter-modulated MLPs are able to preserve a meaningful degree of semantic reasoning capability while retaining episodic information encoded through memory injection.

### 4.2.3. PERSONALIZATION

Following Generative Adapter (Chen et al., 2025), we evaluate conversational question answering based on dialogue

*Table 4.* Model size, memory footprint, and evaluation results on multi-hop and long-context QA benchmarks.

| Model | #Params | Equivalent Storage Size | HotpotQA | NarrativeQA | Qasper | MultiFieldQA$_{\text{EN}}$ | 2WikiMQA | MuSiQue |
|---|---|---|---|---|---|---|---|---|
| Llama3.1-8B-8192 (Meta, 2024) | - | 8192 | 0.431 | 0.243 | 0.300 | 0.432 | 0.349 | 0.250 |
| MemoryLLM-8B-12800 (Wang et al., 2025a) | 40M | 12800 | 0.379 | **0.235** | **0.316** | **0.422** | 0.322 | 0.204 |
| Generative Adapter (Chen et al., 2025) | 500M | 256 | 0.385 | 0.148 | 0.181 | 0.186 | 0.337 | 0.212 |
| HyperMem | 300M | 258 | **0.420** | 0.154 | 0.227 | 0.213 | **0.386** | **0.238** |

*Table 5.* MSC personalized question answering evaluation.

| Model | Equivalent Storage Size | F1 |
|---|---|---|
| Activation Beacon-512 (Zhang et al., 2025) | 512 | 40.8 |
| Activation Beacon-1024 (Zhang et al., 2025) | 1024 | 44.4 |
| Generative Adapter (Chen et al., 2025) | 256 | 40.2 |
| HyperMem | 258 | 43.0 |

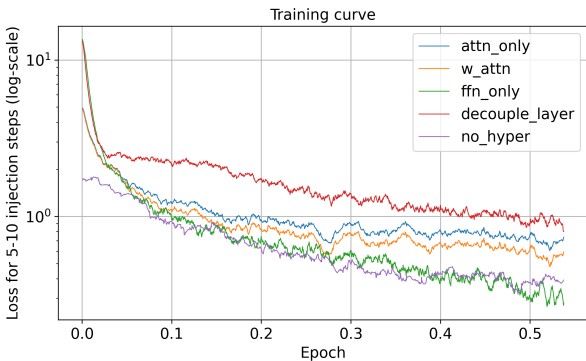

*Figure 4.* **Training curve with different structural choices**. Data points are smoothed using moving average.

history on the MemGPT-MSC (Packer et al., 2023) benchmark, which measures practical utility in personalization scenarios. Experimental results (Table 5) indicate that HyperMem performs on par with related methods, including context-based memory approaches with high compression ratios such as Activation Beacon (Zhang et al., 2025). These results demonstrate that HyperMem can support personalized responses based on installed personas and past episodic events, even when long-term information is stored implicitly in parameters rather than explicitly in context.

### 4.3. Ablation Studies

We conduct ablation studies to explore alternative design choices and to validate the motivations underlying HyperMem.

#### 4.3.1. MLP VS. ATTENTION MODULATION

We investigate whether modulating attention parameters via the hypernetwork can achieve performance comparable to MLP modulation. During post-training, we experiment with three variants: modulating only attention layers (*attn_only*), modulating only FFN layers (*ffn_only*), and modulating both (*w_attn*). Early training curves (loss w.r.t. 5-10 injections) are shown in Figure 4. We find that modulating attention layers or modulating both MLP and attention layers can not yield better performance or convergence speed, compared to modulating only FFN layers. Intuitively, attention mechanisms primarily function as context retrieval modules, while MLP layers encode richer semantic transformations. This observation aligns with findings from prior work on Transformer interpretability (Geva et al., 2021).

#### 4.3.2. HYPERNETWORK DESIGN

In this section, we explore alternative designs for the memory management and hypernetwork components.

**Decoupling from language modeling.** Since LoRA$_{\text{enc}}$ is added to the pre-trained language model parameters while it also interacts intensively with the parameter space, we investigate whether using separate parameters improves performance. To this end, we remove the LoRA$_{\text{enc}}$ and instead employ a randomly initialized single Transformer layer, architecturally identical to LLaMA3.1-8B, as the alternative memory management function. This layer is shared across all Transformer blocks. As shown in Figure 4 (*decouple_layer*), we observe that the convergence becomes significantly slower than the LoRA$_{\text{enc}}$ design.

**Removing the hypernetwork.** We also explore whether removing the hypernetwork entirely and directly using the memory pool as inference-time context could be a more efficient design. In such case, the memory pool act as extended kv cache for attention as vanilla Transformers. This design shares some characteristics with several existing context-based memory approaches (Ge et al., 2023; Zhang et al., 2025). Note that while memory injection becomes faster in this case, generation incurs additional overhead. We denote this design as *no_hyper*. As shown in Figure 4, although this variant is effective and robust, it does not exhibit a clear advantage in episodic memory loss compared to the proposed approach in early training.

### 4.3.3. HYPERPARAMETERS

In this part, we evaluate the impact of key hyperparameters. For both encoder and decoder LoRA modules, we experimented with rank $r = 16$ and $r = 64$, observing significantly better training loss with $r = 64$ ($\Delta loss = 0.1$ for the associative memory task). To scale memory size, we compare $K = 3 \times 86$ with $K = 3 \times 172$. While the larger memory size shows advantages in early training, it necessitates a redesign of gradient propagation settings, making direct comparison unreliable. We leave a systematic study of memory scaling laws to future work. For the scaling coefficient $\alpha$ in Equation (8), we find that $\alpha = 0.01$ yields the best validation loss, whereas larger values lead to slower convergence or instability.

## 5. Related Work

**Memory in Large Language Models.** Memory has emerged as a central theme in the development of next-generation large language models. As surveyed in (Wang et al., 2024c), existing approaches to incorporating memory can be broadly categorized by their storage and access mechanisms, with particular distinctions between context-based memory, parameter-based memory, and fixed-size explicit memory.

*Context-based memory* stores past information directly in the input context, often mediated by external agents or retrieval systems. Representative works include industrial-scale memory agents(Li et al., 2025; Chhikara et al., 2025; Packer et al., 2023), with hand-designed or learned (Wang et al., 2025b) management policies. While these approaches are flexible and effective, their compression rates are typically limited, as the compression strategy often relies on heuristics. As a result, they tend to incur substantial "memory access latency" during inference, especially when long histories are involved.

*Parameter-based memory* encodes information directly into model weights, most commonly through test-time training or online parameter updates (Wang et al., b;a). These methods achieve high compression efficiency and fast reading speed, and benefit from data-driven updates, but suffer from high "writing cost", as each memory update requires gradient-based optimization.

*Fixed-size explicit memory* represents a third line of work, which stores information in a bounded, continuous, and often vectorized external memory (Das et al., 2024; Wang et al., 2024b; Chen et al., 2025). These approaches avoid the overhead of test-time training while maintaining a constant memory footprint. However, designing effective mechanisms for storing, updating, and utilizing purely numerical memory representations remains a long-standing challenge.

Notably, several methods that initially adopt fixed-size memory ultimately reintroduce retrieval mechanisms (Tack et al., 2024; Wang et al., 2025a), effectively reverting to context-based memory formulations. HyperMem can be classified into this category by its nature.

**Context Compression.** Compressing long contexts has long been a major research direction in large language modeling. Mainstream approaches include summarizing long sequences into a small number of representative tokens (Ge et al., 2023; Zhang et al., 2025; Wang et al., 2024b), as well as architectural innovations such as linear-time Transformers and recurrent-style models (Katharopoulos et al., 2020; Sun et al., 2023). These methods are largely orthogonal to our work: HyperMem focuses on transforming contextual information into long-term memory encoded in model parameters, a capability that is absent from most context compression approaches.

**Other Related Work.** Designing neural architectures with built-in memory mechanisms has long been considered a fundamental challenge in machine learning (Behrouz et al., 2024; Behrouz et al.). However, many existing proposals involve complex architectures that are difficult to scale or cannot readily leverage large pretrained language models. Besides, our object-driven, end-to-end approach avoids complex hand-crafted explicit forgetting mechanisms in these proposals. For hypernetwork-based research, (Phang et al., 2023) has used hypernetworks to encode task-specific semantic rather than general-purpose memory, and their architecture significantly differs from ours.

## 6. Conclusion

In this work, we introduced *HyperMem*, a memory framework that unifies episodic and semantic memory in large language models by dynamically transforming contextual episodes into FeedForward layer modulations. HyperMem maintains a fixed-size, recurrent memory that is projected into parameter-shaped updates. Through a two-stage training strategy, HyperMem learns to both reconstruct fine-grained episodic details and abstract semantic knowledge from injected text. Extensive experiments demonstrate that our model exhibits strong episodic memory capabilities, including accurate random-access retrieval and long-range retention across multiple injections, while also preserving competitive semantic reasoning performance on reading comprehension and personalization benchmarks. Overall, HyperMem provides evidence that model parameters can serve as a flexible and expressive medium for long-term memory. We believe this perspective opens new avenues for designing memory-augmented foundation models that are both efficient and scalable in lifelong learning.

## Impact Statement

The data used to train HyperMem is obtained exclusively from publicly available sources or through commercial licenses. No unauthorized or private data has been included. As HyperMem is developed upon a foundation language model and refined through post-training, harmful contents could potentially be elicited from the released model despite the efforts made for safety. The generated contents by our model do not represent the opinions of the authors or entities involved.

## Acknowledgments

This work is supported by the National Science and Technology Major Project (No. 2022ZD0116314) and the National Science Foundation of China (No. 62106249). We would like to thank the colleagues from Beijing Academy of Artificial Intelligence (BAAI) and Spin Matrix for their help on computational resources and experimental devices, and all other colleagues' strong support for this project.

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
