# OpenReview forum: "Integrated Episodic and Semantic Memory via Modulating Transformer FeedForward Layers"
_ICML.cc/2026/Conference — ICML 2026 regular_

### Official Review · Reviewer_29vb · 2026-02-25

**Soundness:** 3
**Presentation:** 3
**Significance:** 3
**Originality:** 4
**Overall Recommendation:** 5
**Confidence:** 3

**Summary:**

The paper introduces HyperMem, a hypernetwork-based framework designed to unify episodic and semantic memory within Large Language Models. Addressing the limitations of context-based memory and parameter-based memory , the authors propose a mechanism where a recurrent, fixed-size memory pool is dynamically transformed into parameter modulations during inference.

**Compliance With Llm Reviewing Policy:**

Affirmed.

**Key Questions For Authors:**

- You use a memory pool of size $K=258$ tokens. While the retention curve shows stability up to 20 injections, is there a theoretical or empirical "collapse point" where earlier memories are catastrophically forgotten? How does the model choose what to overwrite in the recurrent update $M_t = f(c_t, M_{t-1})$? Is this purely implicit via the LoRA encoder training?

- How does HyperMem fundamentally differ in capability from Recurrent Neural Networks or Linear Transformers which also maintain a fixed-size state? Is the key advantage the ability to leverage the pre-trained strong reasoning capabilities of the frozen Llama backbone (which RNNs lack)? A discussion on this distinction would strengthen the positioning.

**Limitations:**

yes

**Strengths And Weaknesses:**

# Soundness: 3 (Good)

## Strengths
- The technical approach is grounded in valid hypotheses regarding the role of FeedForward layers as key-value memories.
- The mechanism of using a hypernetwork to generate LoRA-like updates based on a recurrent state is a technically sound way to bridge the gap between fixed weights and dynamic context.
- The ablation studies are well-conceived, particularly the comparison between modulating MLPs vs. Attention layers, which empirically supports the theoretical intuition that MLPs store semantic knowledge.
- The "retention curve" experiment (Figure 3) provides strong evidence for the efficiency of their recurrent injection compared to MemoryLLM's fixed-window approach.

## Weaknesses
- The definition of "Semantic Memory" in the experiments relies heavily on standard Reading Comprehension. While this proxies for semantic understanding, it doesn't fully prove the memory has been "consolidated" into general knowledge in the same way human semantic memory works.
- The baseline comparisons, while including MemoryLLM and Generative Adapters, lack comparisons to more recent linear-attention or state-space models which also claim to solve the infinite context problem with fixed states, though the paper positions itself specifically against memory-augmented transformers.

# Presentation: 3 (Good)
## Strengths
- The paper is written with high clarity.

## Weaknesses
- The training recipe is somewhat dense. Specifically, the explanation of the "gradient flow" through the recurrent memory steps during training could be clarified. The phrase "limit gradient backpropagation to at most 13 injection steps" needs more detail regarding how this affects long-term dependency learning versus the inference-time behavior.

# Significance: 3 (Good)
## Strengths
- This work addresses a critical bottleneck in AGI: the "vanishing gradient" of lifelong learning.

## Weaknesses
- The scale of experiments (Llama-3.1-8B) is standard for academic papers but leaves open the question of how this hypernetwork approach scales to 70B+ models where the FFNs are significantly larger.

# Originality: 4 (Excellent)

## Strengths
- While hypernetworks and memory mechanisms exist, the specific combination of recurrent memory pools + FFN modulation to unify episodic and semantic functions is novel.
- The conceptual shift from "context as inputs" to "context as parameter modulations" is distinct from standard RAG or long-context Transformers.

---

> ### Author Rebuttal · Authors · 2026-03-30
>
> We sincerely appreciate the reviewer's careful reading and constructive comments.
>
> Weakness 1
>
> We agree that reading comprehension is an imperfect proxy for semantic consolidation. In HyperMem, reading comprehension is evaluated context-free, relying purely on modulated parameters, which partially validates semantic memory at the concept and result level, though not at the behavioral level of human semantic consolidation. We will add discussion acknowledging this limitation and clarifying our claims accordingly.
>
> Weakness 2
>
> SSM/linear-attention models (e.g., Mamba, RWKV) are fundamentally different in scope: they replace the attention mechanism entirely and must be trained from scratch, whereas HyperMem is a plug-in memory module for frozen pretrained LLMs. A direct performance comparison would therefore conflate backbone capability differences with memory mechanism differences. We will clarify this distinction in the "Context Compression" subsection of the related work, which already mentions this line of work.
>
>
> Question 1
>
> There is no explicit overwriting mechanism. Memory updates are learned end-to-end via the LoRA encoder, meaning the model implicitly learns a compression policy rather than applying hand-crafted forgetting rules. A fixed-size memory fundamentally cannot handle unlimited-length sequences and should cooperate with other memory forms in lifelong settings [1]. Preliminary tests on collapse behavior show that with memory size 258, forgetting behavior varies sharply across samples beyond 16k tokens. Memory size scaling is left as future work.
>
> [1] Wang et al., Towards Lifespan Cognitive Systems. arXiv:2409.13265
>
> Question 2
>
> The key distinction is exactly as the reviewer identifies: RNNs and linear Transformers maintain fixed-size states but are trained from scratch, forfeiting the reasoning priors of large pretrained models. HyperMem instead treats the frozen Llama backbone as a fixed reasoning engine and learns only to modulate its FFN parameters, keeping the pretrained capabilities. We will make this distinction explicit as noted above.
>
> Thanks again for your time and efforts in reviewing our work and we are happy to have further discussions.

---

> > ### Author Rebuttal · Reviewer_29vb · 2026-04-04
> >
> > Thank you for your rebuttal. After carefully considering your points as well as the feedback from other reviewers, I believe the score I have given is fair and reasonable.

---

### Official Review · Reviewer_atLS · 2026-03-08

**Soundness:** 2
**Presentation:** 3
**Significance:** 3
**Originality:** 3
**Overall Recommendation:** 3
**Confidence:** 4

**Summary:**

This paper proposes HyperMem, a novel architecture that dynamically converts input context into parameter updates for Transformer feed-forward networks (FFN) via a hypernetwork, aiming to unify episodic memory and semantic memory in large language models. Instead of relying on traditional long-context attention windows, this method compresses an unbounded context stream into fixed-size parameter modulation, enabling efficient long-range information storage and retrieval while significantly reducing inference overhead. Experimental results demonstrate that HyperMem outperforms baseline models such as MemoryLLM and Generative Adapter on raw text retrieval, long-text question answering, and personalized tasks, validating that parametric memory represents a feasible direction for achieving lifelong learning.

**Compliance With Llm Reviewing Policy:**

Affirmed.

**Final Justification:**

The rebuttal helps clarify the authors’ intended meaning of “unified” and acknowledges the current scope limitation. However, my main concerns remain. In particular, the response does not provide new evidence that episodic and semantic memory are truly unified beyond sharing the same mechanism; the semantic-memory evidence is still mixed; and the capacity/scaling concern is explicitly left to future work rather than resolved. These are central to the paper’s main claim and would likely require substantial new experiments or analyses rather than a short rebuttal.

**Key Questions For Authors:**

1. Although the paper claims that the memory pool can be continuously updated recurrently, each layer only has 3×86 slots, which constitutes a severe capacity bottleneck. Meanwhile, the paper only fine-tunes and tests within the 8k–16k token range. For real lifelong learning scenarios or extremely long texts of 100k+ tokens (such as entire books), is this fixed and small parameter-increment space really sufficient to encode and preserve rich details?
2. Can further visualization of FFN weight changes (ΔW) be provided? For example, are these changes generally sparse or dense, and do they exhibit structural patterns across layers or modules? This would help clarify how the model encodes specific facts in the parameter space.

**Limitations:**

yes

**Strengths And Weaknesses:**

### Strengths

1. The paper focuses on the important issue of long-term memory mechanisms and attempts to unify episodic memory and semantic memory, making the research direction novel.
2. The experimental results on episodic retrieval are relatively convincing, especially the outstanding performance in random-access retrieval, indicating that the proposed parametric memory mechanism has considerable potential.

### Weaknesses

1. The current evidence is insufficient to support the core claim of “unifying episodic and semantic memory”. Existing experiments mostly show that the same mechanism can support both types of tasks, but not that they have been truly modeled in a unified way.
2. There are concerns regarding the choice of baselines and comparison settings; the fairness and completeness of the experimental comparisons need further clarification.
3. The results on semantic tasks are mixed, which are not sufficient to support strong conclusions related to semantic memory.

---

> ### Author Rebuttal · Authors · 2026-03-30
>
> We sincerely appreciate the reviewer's careful reading and constructive comments.
>
> Weakness 1:
>
> We would like to clarify that our goal is making the same FFN parameters support both episodic and semantic memory, switchable via prompting alone, without separate storage or architectural branches, as stated in the introduction (lines 69–78). We will explicitly link these expressions to our definition of "unification" and remove related statements that may cause conceptual discomfort.
>
> Weakness 2
>
> To the best of our knowledge, Generative Adapter is the only work featuring a similar concept of directly mapping context to model parameters, and all the three models are trained with episodic-oriented objectives and stages. Please see also our response to reviewer 29vb for relation to SSM-like methods. We welcome specific suggestions regarding the fairness and completeness of our experimental settings.
>
> Weakness 3
>
> Despite subtask variance in the semantic experiments, HyperMem achieves competitive performance with only 258 memory tokens. Note that in Table 4, HyperMem ranks either 1st or 2nd in all the subtasks, and is the best in the same storage size level on all the tasks. This indicates that it handles the semantic problem, outperforming at least one baseline. We present results transparently rather than selectively.
>
> Question 1
>
> Indeed, the 100k+ scenarios exceed current capacity; this is a fundamental tradeoff shared by all fixed-capacity baselines. Nonetheless, Figure 3 shows that Hypermem features better retention efficiency than MemoryLLM with larger storage size, and memory scaling laws are explicitly left as future work (line 396). Clearly, a fixed-size memory can not handle unlimited length, and it should cooperate with other memory forms to handle arbitrary length [1]. We do not claim that we have solved lifelong memory at arbitrary scale in this paper.
>
> [1] Towards lifespan cognitive systems. arXiv preprint arXiv:2409.13265
>
> Question 2
>
> We did some preliminary inspection on a few cases. It shows that in the minority of cases, ΔW matrices are rank-deficient (< 86, varying largely across cases) with rank growing with the number of injections; in the majority of cases they are dense. We tend to believe that simultaneously handling episodic and semantic memory requires more than modifying a few terms in the key-value-like structures hypothesized by prior work.
>
> Thanks again for your time and efforts in reviewing our work and we are happy to have further discussions.

---

> > ### Author Rebuttal · Reviewer_atLS · 2026-04-02
> >
> > The rebuttal helps clarify the authors’ intended meaning of “unified” and acknowledges the current scope limitation. However, my main concerns remain. In particular, the response does not provide new evidence that episodic and semantic memory are truly unified beyond sharing the same mechanism; the semantic-memory evidence is still mixed; and the capacity/scaling concern is explicitly left to future work rather than resolved. These are central to the paper’s main claim and would likely require substantial new experiments or analyses rather than a short rebuttal.

---

> > > ### Author Response · Authors · 2026-04-06
> > >
> > > ### Details on "mixed evidence"
> > >
> > > Thank you for your response and for clarifying your remaining concerns. We want to make sure we fully understand your point about "mixed evidence": could you please describe this concern in more details? As noted, our method shows consistent improvements over the baseline (GA) with equal budgets across all semantic memory test sets. If your concern is that our model underperforms MemoryLLM-12800 on certain (3 of 6) datasets, we would like to clarify that their context window size is **40~50x** our memory size, and further provide the following experimental results with additional analysis.
> > >
> > > ### Additional Results
> > >
> > > Upon closer inspection, we identified an important factor unrelated to semantic memory capability: answer format mismatching. Although we followed the same QA prompts as MemoryLLM, the formatting conventions in our SFT data appear to influence the model's default output behavior, reducing its sensitivity to task-specific formatting instructions. On the other hand, Longbench has default preferences: HotpotQA, 2WikiMQA, and MuSiQue favor **short** answers (one word/phrase), while NarrativeQA, Qasper, and MultiFieldQA-en favor **longer** answers (half sentences), because the ground-truths are annotated as such. Our current SFT formatting conventions make HyperMem perform well in short-answer settings but its potential is not fully unleashed in long-answer ones. This is a gap more attributable to annotation style and evaluation metrics, rather than to underlying semantic memory ability.
> > >
> > > To investigate this, we conducted an ablation study in which we enriched our SFT data with more diverse answer style instructions and the corresponding ground-truth answer behaviors. The updated Table 4 is shown below:
> > > | Model              | Params | Memory |  HotpotQA | NarrativeQA |    Qasper | MultiFieldQA_en |  2WikiMQA |   MuSiQue |
> > > | ------------------ | -----: | ----------: | --------: | ----------: | --------: | --------------: | --------: | --------: |
> > > | Llama3.1-8B-8192   |      - |        8192 |     0.431 |       0.243 |     0.300 |           0.432 |     0.349 |     0.250 |
> > > | MemoryLLM-8B-12800 |    40M |       12800 |     0.379 |   **0.235** | **0.316** |       **0.422** |     0.322 |     0.204 |
> > > | Generative Adapter |   500M |         256 |   *0.385* |       0.148 |     0.181 |           0.186 |   *0.337* |   *0.212* |
> > > | HyperMem-original        |   300M |         258 | **0.420** |     0.154 |   0.227 |         0.213 | 0.386 | 0.238 |
> > > | HyperMem-new        |   300M |         258 | **0.420** |     *0.176* |   *0.274* |         *0.365* | **0.405** | **0.268** |
> > >
> > >
> > > **We observe that correctly following the formatting prompts does not change the relative standings across tasks, but significantly narrows the gap on Qasper and MultiFieldQA-en**. It also yields moderate gains on tasks where HyperMem already successfully handles the formats. This indicates that the intrinsic task variances still remains, but the large performance gaps are more style-related. We will update the experimental section with the analysis above.
> > >
> > > We hope this additional analysis helps address your concerns regarding the mixed evidence, and we welcome any further discussion.

---

### Official Review · Reviewer_YTvb · 2026-03-13

**Soundness:** 4
**Presentation:** 4
**Significance:** 3
**Originality:** 4
**Overall Recommendation:** 5
**Confidence:** 4

**Summary:**

This paper proposes HyperMem, a hypernetwork-based framework that directly encodes context into model parameters. It introduces a recurrent, fixed-size memory pool at each layer of the Transformer architecture and a hypernetwork which maps this memory  to low-rank modulations of models feedforward (FF) weights. Using this mechanism, the same parameters can support both the recall of textual events (episodic memory) and the parameter-encoded knowledge for general reasoning and language modeling purposes (semantic memory).

The model is post-trained, with two complementary objectives (random-access associative recall and continuation-based generation) and then fine-tuned on long context comprehension tasks. Episodic memory experiments demonstrate that HyperMem achieves strong performance on retrieval and long-range retention tasks but for smaller memory storage than baselines (like MemoryLLM). In semantic memory experiments, HyperMem remains competitive on benchmarks including LongBench and personalization. Then ablation studies revisit the criticality in performance of various architectural components in HyperMem as well as the impact of varying parameters like memory pool sizes, rank in LoRA encoders/decoders or the value of the multiplier in the additive modulation (as in Equation (8)).

**Compliance With Llm Reviewing Policy:**

Affirmed.

**Final Justification:**

This paper proposes an interesting novel direction for unifying episodic and semantic memory: I am maintaining my original positive score, based on my original reviews and author's reactions during the rebuttal stage.

**Key Questions For Authors:**

## Questions

1. It would be great to offer some insights on what some of the potential failure cases would look like: what are some of the cases where this unification might fail (for example when semantic and episodic memory modes/objectives interfere)?

**Strengths And Weaknesses:**

## Strengths

- The paper explores a novel direction for unifying episodic and semantic memory: the same FF-layer parameters can be function both as episodic and semantic memory (with the introduction of a hypernetwork that performs the modulation of FF network weights). This is an interesting approach that builds upon and extends the current consensus that semantic memory signals originate in FFN layers while those for episodic memory come from the context as processed in KV cache.
- HyperMem uses small, fixed-size memory and avoids quadratic attention costs and test-time training, still delivering superior or comparable performance to baseline methods.
- Experiments are comprehensive: for both episodic and semantic memory tasks and with ablation studies covering both architectural choices/components and key hyperparameters.

## Weaknesses

- The approach introduces multiple components: layer-wise recurrent memory pools, encoder/decoder LoRA modules, a hypernetwork and staged training objectives. This complexity could be hard to reproduce and most importantly to adopt in other backbone models, beyond Llama-3.1-8B. Similarly it is not fully clear how the sensitivity of performance to hyperparameter choices will be affected in this case.

---

> ### Author Rebuttal · Authors · 2026-03-30
>
> We sincerely appreciate the reviewer's careful reading and constructive comments.
>
> Weakness:
>
> We appreciate this concern and would like to clarify that HyperMem is no more complex compared to the baselines in terms of overall system complexity. Compared to MemoryLLM-8B, we follow their design of encoder/decoder LoRA modules. For structure, we only add a single linear projection (the hypernetwork), whereas MemoryLLM requires a sophisticated memory dropping mechanism and at least two (usually three) training stages. Compared to Generative Adapter, we avoid matrix factorization, which introduces additional numerical instability. Hyperparameter sensitivity of Hypermem is also mild at least for Llama-3.1-8B backbone: varying the episodic-to-semantic loss ratio within [0.2:0.8, 1:1] does not significantly affect training dynamics (see also Question 1), and learning rates ranging from 0.3× to 3× all result in successful training.
>
> On adaptability to other backbone models: we acknowledge that a definitive answer requires empirical results beyond Llama-3.1-8B, which we plan to add as resources allow. That said, HyperMem only requires standard FFN up/down/gate projections, a structural design shared by a large family of Llama-style models. We expect positive transfer to this family of models, just like MemoryLLM is effective with both Llama2-7B and Llama3-8B backbones without heavy modifications.
>
> Question 1:
>
> This is an insightful question. The two modes are unlikely to interfere at the instruction level, as they are triggered by distinct prompts: spatiotemporal cues for episodic recall vs. open-ended generation for semantic memory. Potential interference is instead at the capacity level, since both objectives share the same fixed-size parameter modulation budget. Empirically, when the episodic loss weight falls below 0.2× the semantic loss, episodic memory fails to converge. Beyond this threshold (episodic:semantic ∈ [0.2:0.8, 1:1]), training dynamics and final performance are largely unaffected, suggesting the two objectives are more complementary than conflicting. We will include this analysis in the revised paper.
>
> Thanks again for your time and efforts in reviewing our work and we are happy to have further discussions.

---

> > ### Author Rebuttal · Reviewer_YTvb · 2026-04-04
> >
> > The authors have successfully addressed my concerns, further strengthening my positive assessment. As such, I am pleased to keep my original favorable score.

---

### Official Review · Reviewer_wk9h · 2026-03-16

**Soundness:** 3
**Presentation:** 3
**Significance:** 3
**Originality:** 3
**Overall Recommendation:** 4
**Confidence:** 2

**Summary:**

This paper introduces Hypermem, which is a hypernetwork based approach that recurrently maps contexts into targeted updates of FFNs in Transformer architectures. Based on this formulation, authors presented a post-train method on the hypernetwork, showing promising results in long-context QA, retrieval, and personalization benchmarks.

**Compliance With Llm Reviewing Policy:**

Affirmed.

**Final Justification:**

The paper is well-written and well-motivated. The experiments support the claims.

**Key Questions For Authors:**

Please see weaknesses.

**Limitations:**

Yes.

**Strengths And Weaknesses:**

- I generally find the paper easy to follow and well-written.
- The presented method is well-motivated and show clear advantage compared to the baselines.

Weaknesses:
- Generally the presented approach includes multiple components, which can be challenging in real life.

---

> ### Author Rebuttal · Authors · 2026-03-30
>
> We thank the reviewer for positive assessment of our work. We are happy to address any questions or clarifications that may arise during the discussion period.

---

> > ### Author Rebuttal · Reviewer_wk9h · 2026-04-06
> >
> > I thank the authors for their effort. I do not have any concern.

---

### Decision · Program_Chairs · 2026-04-30

**Decision:**

Accept (regular)

**Comment:**

The paper proposes HyperMem, a network that encodes context into model parameters. It introduces a recurrent, fixed-size memory pool at each layer of the Transformer architecture and a hypernetwork which maps this memory to low-rank modulations of models feedforward (FF) weights. Using this mechanism, the same parameters can support both the recall of textual events (episodic memory) and the parameter-encoded knowledge for general reasoning and language modeling purposes (semantic memory).

This is a borderline paper that received mixture of positive and negative reviews. On the positive side, reviewers acknowledge novelty and comprehensive experiments. On the negative side, reviewers believe that the proposed approach is too complex - the network has multiple components: layer-wise recurrent memory pools, encoder/decoder LoRA modules, a hypernetwork and staged training objectives. The combination of these components makes it hard to attribute results to each individual choice and apply this idea to other backbone models.

Overall, potential benefits of presenting this work at ICML seem to outweigh reasons to reject. Therefore, I recommend to accept.